# EB1 Restricts Breast Cancer Cell Invadopodia Formation and Matrix Proteolysis via FAK

**DOI:** 10.3390/cells10020388

**Published:** 2021-02-13

**Authors:** Brice Chanez, Kevin Ostacolo, Ali Badache, Sylvie Thuault

**Affiliations:** Centre de Recherche en Cancérologie de Marseille (CRCM), INSERM, Institut Paoli-Calmettes, Aix-Marseille University, CNRS, 13009 Marseille, France; chanezb@ipc.unicancer.fr (B.C.); kja11@hi.is (K.O.)

**Keywords:** breast cancer, invadopodia, microtubules, +TIPs, FAK, extracellular matrix

## Abstract

Regulation of microtubule dynamics by plus-end tracking proteins (+TIPs) plays an essential role in cancer cell migration. However, the role of +TIPs in cancer cell invasion has been poorly addressed. Invadopodia, actin-rich protrusions specialized in extracellular matrix degradation, are essential for cancer cell invasion and metastasis, the leading cause of death in breast cancer. We, therefore, investigated the role of the End Binding protein, EB1, a major hub of the +TIP network, in invadopodia functions. EB1 silencing increased matrix degradation by breast cancer cells. This was recapitulated by depletion of two additional +TIPs and EB1 partners, APC and ACF7, but not by the knockdown of other +TIPs, such as CLASP1/2 or CLIP170. The knockdown of Focal Adhesion Kinase (FAK) was previously proposed to similarly promote invadopodia formation as a consequence of a switch of the Src kinase from focal adhesions to invadopodia. Interestingly, EB1-, APC-, or ACF7-depleted cells had decreased expression/activation of FAK. Remarkably, overexpression of wild type FAK, but not of FAK mutated to prevent Src recruitment, prevented the increased degradative activity induced by EB1 depletion. Overall, we propose that EB1 restricts invadopodia formation through the control of FAK and, consequently, the spatial regulation of Src activity.

## 1. Introduction

Metastatic progression is the leading cause of mortality among breast cancer patients. In order to leave the primary tumor and reach distant organs, cancer cells need to migrate and degrade the surrounding extracellular matrix (ECM). Invadopodia, which are dynamic actin-rich membrane protrusions developed by cancer cells, play a major role in ECM proteolysis [1,2,3]. Invadopodia formation is commonly initiated following Src kinase activation downstream of ligand-induced activation of receptor tyrosine kinases, such as EGF receptor. This leads to the recruitment and activation of signaling and structural factors implicated in actin reorganization, such as Cortactin, Arp2/3, N-WASP, Tks4/5, and the release of proteases such as MT1-MMP, involved in matrix proteolysis. Thus invadopodia appear as key structures involved in cancer cell invasion and metastatic progression [4,5].

Efficient invasion of cancer cells not only relies on ECM degradative potential but also on efficient cell migration. Focal adhesions (FAs), which are integrin-linked protein complexes acting as mechanical linkers between the ECM and the actin cytoskeleton, play a major role in cell spreading and migration [6,7]. Interestingly, invadopodia and FAs share many molecular components, including adaptor proteins such as paxillin and talin, actin regulators such as cofilin and VASP, and signaling enzymes such as Src. Previous studies have suggested that tight control of the dynamics of FAs and of invadopodia is crucial for efficient cell invasion. As an example, the Focal Adhesion Kinase (FAK), whose activity is crucial for FA turnover, has been reported to inhibit ECM proteolysis [8,9,10,11]. Indeed, upon FAK depletion, the Src kinase, critical for invadopodia initiation, and other FA-recruited tyrosine phosphorylated proteins, were released from FAs, allowing them to induce invadopodia formation. Thus, FAK depletion increased cell degradative potential, but not overall cell invasion, since it was also correlated with a defect in cell migration [9,10,11].

While the actin cytoskeleton is a major player in the regulation of the assembly and dynamics of invadopodia and FAs, microtubules (MTs) and intermediate filaments play more specific roles [12,13,14]. MTs are involved in the turnover of FAs. Indeed, MT depolymerization by nocodazole treatment increases the number and size of FAs [15]. Furthermore, regrowth of MTs, following nocodazole washout, induces FA complexes disassembly [16,17]. MTs control FA turnover through the regulation of vesicular trafficking of integrins and other factors to and from FAs, and the regulation of signaling pathways, in particular Rho GTPases activity, governing actin cytoskeleton organization [18,19,20]. MTs are dynamic polymers whose assembly and stability are under the control of MT-associated proteins. Among these, plus-end tracking proteins (+TIPs) bind and concentrate at MT plus-ends [21,22]. +TIPs interact with MT ends and with each other in order to generate a complex protein network that controls MT dynamics. The End Binding (EB) family of proteins has emerged as a central “hub” in the formation of this network [21]. EB1-3 interacts directly with tubulin via their N-terminal Calponin Homology (CH) domain. Their C-terminal EB Homology (EBH) domain allows their dimerization through the formation of a coiled coil bundle. The helix bundle of the EBH domain dimer forms a hydrophobic groove, which allows EBs to interact with a variety of proteins containing a SxIP sequence motif, including APC, ACF7/MACF1, and CLASP1/2. The acidic tail of EBs mediates the binding of cytoskeleton-associated protein-glycine-rich (CAP-Gly) motifs containing +TIPs, such as CLIP170 and p150glued. The regulation of MT dynamics by +TIPs is essential for directional cell migration [23,24]. MT ends are found in close proximity to FAs where they interact via +TIPs with components of cortical MT stabilizing complexes in order to be captured and stabilized at the cell cortex of the leading edge of migrating cells [13]. Moreover, +TIPs were implicated in FA turnover, inhibition of their activity being generally correlated with stabilization of FAs [20,25,26,27,28].

While the contribution of MTs and +TIPs in FA dynamics are well documented, much less is known concerning their potential function in invadopodia regulation and ECM degradation. We, therefore, investigated the impact of EB1, the central hub in the +TIP network, on invadopodia dependent ECM degradation in breast cancer cell models. We observed that EB1 silencing enhanced the degradation of a reconstituted matrix, by increasing the number of invadopodia. Depletion of the APC and ACF7 +TIPs led to a similar phenotype, whereas depletion of CLASP1/2 and CLIP170 had no impact on invadopodia. Analysis of Src and FAK activities suggest that EB1, APC, and ACF7 control the balance between FAs and invadopodia by promoting FAK activation, favoring Src localization at FAs, and thus negatively impacting invadopodia formation.

## 2. Materials and Methods

### 2.1. Cell Lines

Human MDA-MB-231 breast cancer cells (kindly provided by O. Segatto, IRCCS, Roma) were cultured in Dulbecco’s Modified Eagle Medium (DMEM) (ThermoFisher, Waltham, MA, USA), supplemented with 10% fetal bovine serum (Eurobio, Les Ullis, France). Normal human MCF10A breast epithelial cells (ATCC/LGC Standards) were cultured in DMEM/F12 (ThermoFisher) supplemented with 5% fetal bovine serum, 20 ng/mL EGF (Peprotech, Neuilly sur Seine, France), 0.5 µg/mL hydrocortisone (Sigma-Aldrich, Saint-Louis, MO, USA), 100 ng/mL cholera toxin (Sigma-Aldrich), and 10 µL/mL insulin (Sigma-Aldrich). Cells were cultured at 37 °C in a humidified atmosphere with 5% CO_2_ and checked regularly for mycoplasma contamination. Stable MDA-MB-231 cell lines were generated by transfection of the mCherry plasmid constructs by Lipofectamine 2000 (Invitrogen, ThermoFisher), selection with 1 mg/mL geneticin, and sorting by flow cytometry to select cell populations expressing moderate levels of the transgenes. For TGF-β-induced EMT, MCF10A cells were treated for 6 days with 10 ng/mL human recombinant TGF-β derived from HEK293 cells (Peprotech). GM6001 was purchased from Calbiochem (Sigma-Aldrich). Paclitaxel and Nocodazole were purchased from Sigma-Aldrich.

### 2.2. Sequences of siRNAs and Transfection

SiRNAs were home-designed and purchased from ThermoFisher. The siRNAs targeting human EB1 (siEB1_1: NM_012325-1262), APC (NM_000038-4610 or -9773), ACF7 (NM_012090-3015), CLASP1 (NM_015282-709), CLASP2 (NM_015097-1294), and LacZ (M55068-427) used as a negative control, have been previously described in Zaoui et al., as well as their efficiency and specificity (Appendix A) [29]. The siRNA sequence against CLIP170 (NM_002956) and the second siRNA sequence against EB1 (siEB1_2: NM_012325-1221) are indicated in Appendix A. Cells were transfected with siRNA at a final concentration of 30 nM using Lipofectamine RNAiMAX reagent (ThermoFisher) according to the manufacturer’s instructions. For MDA-MB-231 cells, functional assays and assessment of knockdown efficiency were performed 3 days after transfection. For rescue experiments, MDA-MB-231 cells were transfected with expression plasmids using Amaxa nucleofection technology (Lonza, Basel, Switzerland) one day after siRNA transfection. For TGF-β-transformed MCF10A cells, cells were treated for 5 days with TGF-β before the day of transfection. At the day of transfection, cells were treated a 2nd time with TGF-β for 1 additional day.

### 2.3. Plasmid Constructs

Wild type FAK and FAK Y397F were amplified by PCR from pAcGFP1-Hyg-C1-FAK wild type and Y397F (a kind gift from P. Rondé, LBP, Strasbourg) with appropriate primers (Appendix A). PCR products were used to generate pDONR-FAK and pDONR-FAK Y397F, respectively, before subcloning in the destination vector pDEST-mCherry C1 (a gift from F. Lembo, CRCM) by Gateway technology, to obtain FAK and FAK Y397F fused to the C-terminus of mCherry. Wild type EB1 was amplified by PCR from peGFP-N1-EB1 (#39299; Addgene, a gift from T. Mitchison and J. Tirnauer, Harvard) with appropriate primers (Appendix A). PCR products were used to generate pDONR-EB1 before subcloning in the destination vector pDEST-mCherry N1 (#31907, Addgene, a gift from R.M. Shaw, San Francisco, CA, USA). All constructs were sequence verified.

### 2.4. Antibodies

Rabbit antibodies against Tks5 (M-300), CLASP2 (H-40) and CLASP1 (H-70), mouse antibody against CLIP170 (F3) and rat antibody against EB1 (KT51) were purchased from Santa-Cruz Biotechnology, Dallas, TX, US. Mouse antibody against Cortactin (clone 4F11) was purchased from Millipore. Rabbit antibody against mCherry was from Abcam, Cambridge, United Kingdom. Rabbit antibody against Phospho-FAK (Tyr397) was from ThermoFisher. Mouse antibodies against EB1 and Src (L4A1) and rabbit antibody against p-Y416-Src were from Cell Signaling Technology, Danvers, MA, USA. Anti-α-Tubulin (mouse, clone DM1A) was from Sigma-Aldrich. Rabbit antibody against FAK (AHO0502) was from BioSource, ThermoFisher.

### 2.5. Western Blotting

Protein extracts were prepared, quantified, and denatured in SDS loading buffer before running the samples on Novex NuPAGE Bis-Tris 4–12% gels using a MOPS based running buffer (Life Technologies, Carlsbad, CA, USA). Proteins were then transferred onto nitrocellulose membranes. Membranes were incubated with primary antibodies followed by incubation with secondary antibodies coupled to HRP. Antibodies were detected by chemoluminescence. Signal quantification was performed using the Gel analyzing tool from Image J 1.53c.

### 2.6. Matrix Degradation Assay and Immunolabeling

Coverslips coated with Oregon Green 488-conjugated gelatin (Life technologies, ThermoFisher) were prepared as described in Thuault et al. [30]. After seeding for 4 h on gelatin-coated coverslips rehydrated in complete growth medium for 1 h before use, cells were fixed with a solution of 4% formaldehyde in PBS, permeabilized with 0.1% Triton X-100, and blocked with 1% BSA. When EB1 was labeled, cells were fixed with methanol before a 2nd fixation step with 4% formaldehyde in PBS and blocking with 1% BSA. Antibodies directed against the target proteins and secondary antibodies labeled with DyLight 405 or AlexaFluor 594 (Jackson ImmunoResearch, West Grove, PA, US) were then used for immunolabeling. A Zeiss structured light ApoTome microscope equipped with a 63× 1.4 plan ApoChromat objective was used for image acquisition (Zeiss, Jena, Germany). Ten random fields per coverslips, using 2 coverslips per condition per experiment, were imaged to assess the percentage of cells degrading. A home-made Fiji macro was used to analyze matrix degradation. The mean degraded area and the number of degradation foci per cell were quantified for 25 cells per condition per experiment.

### 2.7. Statistical Analysis

GraphPad Prism software was used to perform all statistical analyses. To determine significant differences between data groups, the paired *t*-test, the unpaired one-tailed *t*-test, with Welch correction, and the Mann–Whitney test was used. Graphs were plotted using Prism to show the mean and SEM. The mean of each individual experiment was also reported. *p*-values were indicated on the graph as * *p* < 0.05, ** *p* < 0.01, *** *p* < 0.001.

## 3. Results and Discussion

### 3.1. EB1 Restricts ECM Degradation via Inhibition of Invadopodia Formation in Breast Cancer Cell Lines

We investigated the contribution of EB1 to breast cancer cells ability to form invadopodia and degrade the ECM. For that purpose, invasive MDA-MB-231 breast cancer cells transfected with either a control siRNA (siLacZ) or siRNAs directed against EB1 were seeded on an artificial ECM composed of fluorescently-labeled gelatin for 4 h (Figure 1A). Invadopodia were identified by co-labeling of Cortactin and TKS5, two constitutive components of invadopodia [1]; degraded ECM appeared as dark, non-fluorescent spots. The two distinct siRNA sequences used to target EB1 efficiently decreased EB1 protein levels (Appendix A). Silencing of EB1 had modest or no impact on the percentage of cells degrading the ECM (Figure 1B). Interestingly, depletion of EB1 induced an increase in ECM degradation per cell (Figure 1C). To verify that ECM proteolysis actually involved matrix metalloproteases (MMP) activity, we treated the cells with the general inhibitor of MMP, GM6001. The treatment abolished ECM degradation induced by control cells as well as EB1 depleted cells (Appendix A). Increased ECM degradation in EB1-depleted cells was not the consequence of enlarged degradation foci (Figure 1D) but of a greater number of degradation foci per cell (Figure 1E). To further verify that the observed effects were not the consequence of siRNA off-targets, we restored EB1 expression by co-transfecting EB1 fused to mCherry (which is resistant to siEB1_2 that targets the 3′UTR sequence of endogenous EB1). Re-expression of EB1 reverted the increased degradative phenotype, bringing it back to control cell degradation levels (Appendix A).

To strengthen our observations, we investigated the impact of EB1 depletion on the degradative potential of another cellular model, MCF10A normal breast epithelial cells that had undergone epithelial to mesenchymal transition (EMT) following TGF-β treatment, a process inducing migratory and invasive properties. As previously described [31], normal MCF10A cells poorly degraded the matrix (Figure 2A). However, upon TGF-β-induced EMT, their degradative potential was increased (Figure 2A). Upon depletion of EB1 (Appendix A), a further increase in the total degraded area per cell as well as in the number of degradation foci was observed (Figure 2A,C,E). However, as in the MDA-MB-231 cell model, EB1 depletion did not affect the percentage of degrading cells (Figure 2B) nor the size of degradation foci (Figure 2D).

Overall, these experiments demonstrate that EB1 inhibits ECM proteolysis by restraining invadopodia formation in breast cancer cells. This is in contrast to osteoclasts, in which EB1 is required for podosome belt formation, podosomes being degradative machinery formed by some normal cells, sharing similarities, but also major differences, with invadopodia [2,32]. However, EB1 has been described to be enriched in podosomes, whereas we could not detect EB1 at invadopodia (data not shown), which suggests that the differential impact of EB1 towards invadopodia and podosomes is linked to different modes of action.

### 3.2. APC and ACF7, Similarly to EB1, Restrict ECM Proteolysis by Restraining Invadopodia Formation

EB1 interacts with SxIP-containing proteins, such as CLASP1/2, APC, and ACF7, and CAP-Gly-containing proteins, such as CLIP170 and p150glued in order to regulate the dynamics and stability of MTs [21,22]. We, therefore, investigated whether EB1 binding partners were also involved in matrix proteolysis and invadopodia regulation. MDA-MB-231 cells were transfected with validated siRNAs [29] specifically targeting APC, ACF7, CLASP1, CLASP2, or CLIP170 before seeding on fluorescently-labeled gelatin (Figure 3A). As observed for EB1, APC, CLASP1, CLASP2, or CLIP170, silencing did not affect the percentage of cells degrading the ECM (Figure 3B); only ACF7 silencing significantly increased the percentage of degrading cells. Furthermore, we observed that APC or ACF7 depletion, similarly to EB1 depletion, increased the degradative potential of MDA-MB-231 cells. Depletion of CLASP1, CLASP2, or CLIP170 had no impact (Figure 3C); we verified by Western blotting that this was not due to a lack of efficiency of CLASP1, CLASP2, or CLIP170 knock-down (Appendix A). Similarly to EB1-depleted cells, the increased degradative potential of APC- and ACF7- depleted cells was correlated to an increased number of degradation foci, but not foci size (Figure 3D,E).

Our results uncovered three novel negative regulators of invadopodia belonging to the family of +TIPs factors: EB1, APC, and ACF7. The fact that APC and ACF7 silencing leads to strikingly similar effects to EB1 knockdown suggests that APC and ACF7 might function in collaboration with EB1 to restrain invadopodia formation. EB1, APC, ACF7, CLASP1/2, and CLIP170 have all been implicated in migration-promoting pathways [25,33,34,35,36,37]. Our results show, however, that they have distinct roles in invadopodia formation and ECM degradation. A previous study implicated CLASP2 in the control of ECM degradation at FAs [27], but its impact on ventral ECM degradation through invadopodia has not been addressed. Our observations suggest that CLASP2 differentially regulates ECM degradation at FAs and at invadopodia.

Currently, few proteins have been reported as negative regulators of invadopodia function. Among them, the Rho GTPase RhoG and its guanine exchange factor SGEF, through the regulation of paxillin phosphorylation, and Ezrin, which promotes Calpain protease activity, favor invadopodia disassembly and thereby restrain ECM degradation [38,39,40]. FAK and Laminin-332 have also been reported as negative regulators of invadopodia. Cells depleted of Laminin-332 harboring decreased level of active FAK, and its effects might be mediated by FAK downregulation [41]. FAK is critical for the localization of active Src to FAs. Several studies indicate that depletion of FAK induces a redistribution of active Src from FAs to invadopodia [9,10,11]. Although depletion of these factors favors ECM proteolysis, cells depleted of these factors have an impaired global invasive potential, related to impaired cell migration, implying that efficient cell invasion requires timely regulation of migration and ECM proteolysis.

### 3.3. FAK Is Necessary for EB1 to Restrict Invadopodia Formation

Src is a critical element of invadopodia formation [1,42]. We, therefore, analyzed how EB1, APC, and ACF7 silencing affected Src expression and activity in invadopodia-forming cells. Total Src levels were similar to control in EB1-, APC-, and ACF7-depleted cells (Figure 4A). However, we observed a two-fold reduction of activated Src (p-Y416-Src) in EB1-, APC-, or ACF7-depleted cells compared to control cells (Figure 4A). Thus, the observed increase in invadopodia formation and ECM degradation was surprisingly associated with decreased Src activity. This observation suggested that EB1, APC, and ACF7, rather than controlling Src activity, could spatially regulate its availability, as described before upon FAK depletion [9,10,11]. +TIPs being important for FAs turnover [20,25,26], we hypothesized that EB1, APC, and ACF7 could act as negative regulators of invadopodia formation through the control of FAK activation. To address this hypothesis, we analyzed FAK levels and activation in EB1-, APC-, and ACF7-depleted invadopodia-forming MDA-MB-231 cells. We observed that total FAK levels were significantly down-regulated in EB1- and APC-depleted cells compared to control cells (Figure 4B). Silencing of ACF7 affected total FAK levels in a less reproducible manner. Following its recruitment to integrin-associated complexes, FAK was auto-phosphorylated on Tyr397. This auto-phosphorylation site is an important SH2 domain binding motif required for Src recruitment and further FAK activation [43]. We observed that p-Y397-FAK levels were strongly down-regulated in EB1-, APC-, and ACF7-depleted cells (79.4 ± 3.7% in EB1, 97.4 ± 0.8% in APC, and 61.4 ± 6.1% in ACF7-depleted cells, respectively). A similar decrease in the levels of p-Y397-FAK has been reported as a consequence of APC depletion in U2OS cells [26].

The decreased activation of FAK in EB1-, APC-, and ACF7-depleted cells could, therefore, be the cause of enhanced invadopodia formation in our cellular model. In order to explore this hypothesis, we evaluated the impact of EB1 silencing on invadopodia formation upon restoration of normal FAK expression levels and/or activity. We generated and selected MDA-MB-231 cells stably expressing wild type FAK fused to mCherry, cells expressing a mutant form of FAK that cannot be phosphorylated on Tyr397 (FAK Y397F), and cells expressing a control vector (mCherry) (Figure 4C). We analyzed the degradative capacity of the three cell lines upon EB1 silencing (Figure 4D). In control mCherry expressing cells, EB1 depletion induced increased ECM degradation and a higher number of degradation foci without impacting the percentage of degrading cells (Figure 4D–G), similarly to what was observed in non-transfected cells (Figure 1). As previously described [9,10,11], we observed that FAK inhibited cells degradative capacity. Indeed, the expression of FAK reduced the percentage of cells degrading while having a minor repressive impact on the total degraded area per cell or on the number of degradation foci per cell (Appendix A). Interestingly, in cells in which FAK expression levels were restored, EB1 depletion did not lead to increased ECM proteolysis or more invadopodia (Figure 4D,F,G). This showed that the effect of EB1 silencing on invadopodia was dependent on the downregulation of FAK. Interestingly, when FAK was mutated to prevent recruitment of SH2 domain-harboring proteins, it was ineffective at reverting the effects of EB1 silencing on MDA-MB-231 degradative activity (Figure 4D,F,G). The rescue of EB1-silenced cells by wild type FAK, but not by a form of FAK unable to recruit Src, strongly suggest that the level of FAK activation, and thus of Src recruitment to FAK, is the crucial event in EB1-mediated invadopodia regulation. Molecular mechanisms regulating FAK autophosphorylation are diverse, including, but not limited to, interactions with inhibitory factors, inhibitory phosphorylation or dephosphorylation by phosphatases [44]. Further investigations are required to understand how FAK levels and activation are regulated by EB1, APC, and ACF7.

Considering the inter-connection between +TIPs and the MT cytoskeleton, our observation that depletion of EB1, APC, or ACF7 increases ECM degradation appears in contradiction with previous reports indicating that invadopodia, in similar experimental settings, were insensitive to MT-stabilizing or -destabilizing agents in breast cancer cells [12,45]. However, it should be mentioned that MTs are required for invadopodia formation in melanoma cancer cells [46,47], and for podosomes belt formation [31,48,49,50], suggesting that MT implication in invadosomes is cell type dependent. We verified the contribution of MTs in invadopodia formation and ECM degradation in our cellular model by treating MDA-MB-231 cells with the stabilizing agent paclitaxel or destabilizing agent nocodazole at doses that globally disturb MT network (Appendix A). The efficacy of the treatments was validated by analyzing EB1 comets and overall MT cytoskeleton organization; EB1 comets were greatly impaired, and MT cytoskeleton appears as bundles or globally depolymerized, respectively, after paclitaxel or nocodazole treatment (Appendix A). As previously reported, such treatments had no impact on invadopodia and ECM proteolysis (Appendix A). In light of our observations that regulation of FAK activity is important for EB1 control of invadopodia function, we evaluated the impact of these MT targeting agents (MTAs) on FAK levels and activation in MDA-MB-231 cells after 4 h of adhesion to gelatin-coated plates. We observed that neither FAK levels nor FAK activation was impacted by MTA treatments (Appendix A). The fact that MTAs, in contrast to EB1 silencing, do not impact FAK activity could explain why MT cytoskeleton disruption and EB1 silencing differentially impact invadopodia and cell ECM proteolytic activity. Furthermore, these observations implied that the observed phenotype of EB1-silenced cells might be a consequence of the long-term effect of EB1 depletion (72 h), potentially acting at the transcriptional level, the effect that cannot be resumed by short-term MTA treatment.

Furthermore, we identified that the +TIPs EB1, APC, and ACF7 are negative regulators of invadopodia formation and function, whereas CLASP1/2 and CLIP170 are not. Globally these factors are required for persistent directional cell migration [25,34,35,36,37,51]. Their differential impact on ECM proteolysis is, therefore, appealing. A property that distinguishes APC and ACF7 from the other two +TIPs, is their ability to not only promote MT growth but also to interact with the actin cytoskeleton. It would be of interest to explore if the ability of ACF7 and APC to bind actin filaments near or at FAs [25,26] is important for the regulation of FAK activity. Moreover, considering the functional link between EB1, APC, and ACF7 in the control of MT capture in the protrusion of migratory cells and directional migration, analyzing whether these +TIPs act in parallel or in concert through a linear pathway to regulate ECM proteolysis via the control of FAK, would be compelling.

## 4. Conclusions

In conclusion, we demonstrate that EB1 restricts ECM degradation via the regulation of FAK activity. Previous studies show that EB1 is required for persistent directional migration [33,52]. Although not addressed directly in this study, EB1 has been reported to be overexpressed in diverse cancer types and to have pro-invasive functions [53,54,55,56]. Migration and invasion through basal membranes are two distinct sequential steps of the metastatic process. We can envision that in migrating cells, the EB1-FAK connection promotes directed motility and limits the formation of ventral invadopodia. For cells to extravagate through a basal membrane, they would have to interrupt pro-migratory signaling; disruption of the EB1-FAK connection would favor the formation of invadopodia on the cell ventral side, localized degradation of the matrix, and invasion through the basal membrane. Our observations support previous studies suggesting that tight regulation of directional migration and of invadopodia formation is necessary for efficient cell invasion and metastasis.

## Figures and Tables

**Figure 1 cells-10-00388-f001:**
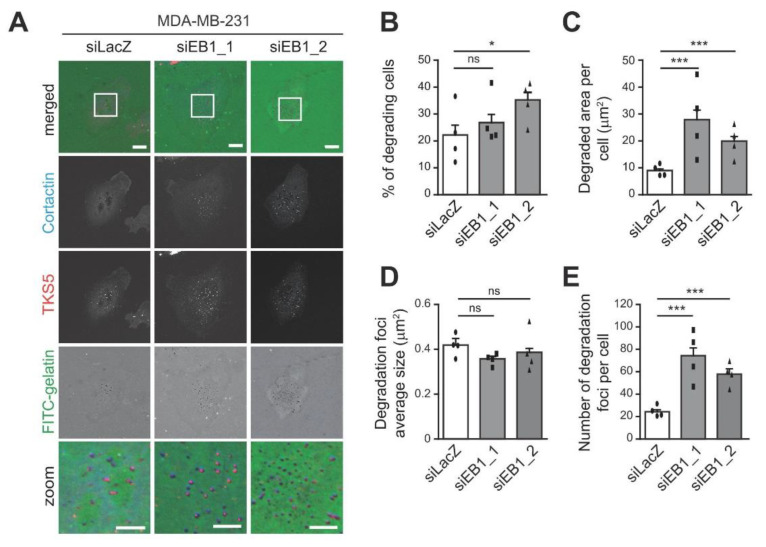
End Binding protein (EB1) restricts extracellular matrix (ECM) degradation via inhibition of invadopodia formation in MDA-MB-231 breast cancer cells. MDA-MB-231 cells were transfected with a control siRNA (siLacZ) or siRNAs against EB1 (siEB1_1 or siEB1_2) and seeded on fluorescently-labeled gelatin (FITC-gelatin) for 4 h. Cells were fixed and stained with antibodies directed against Cortactin and TKS5 to identify invadopodia. Matrix degradation was identified thanks to the appearance of dark spots in FITC-gelatin. (**A**) Representative images are shown. The white-boxed regions are enlarged at the bottom (zoom). Scale bars represent 10 μm in non-enlarged images, 5 μm in enlarged images. (**B**–**E**) The ability of MDA-MB-231 cells to degrade fluorescently-labeled gelatin was analyzed. The percentage of degrading cells (**B**), the degraded area per cell (**C**), the average size of degradation foci (**D**), and the number of degradation foci (**E**) are represented as the mean ± SEM of four independent experiments. The mean of each individual experiment is reported. Percentage of cells degrading was assessed by imaging 10 random fields per coverslip, 2 coverslips per condition per experiment. The unpaired one-tailed *t*-test, with Welch correction, was used to determine significant differences. Matrix degradation was analyzed by quantifying the mean degraded area and the number of degradation foci per cell using home-made Fiji macro, analyzing 25 cells per condition per experiment. The Mann–Whitney test was used to determine significant differences. *** *p* ≤ 0.001, * *p* ≤ 0.05, ns not significant.

**Figure 2 cells-10-00388-f002:**
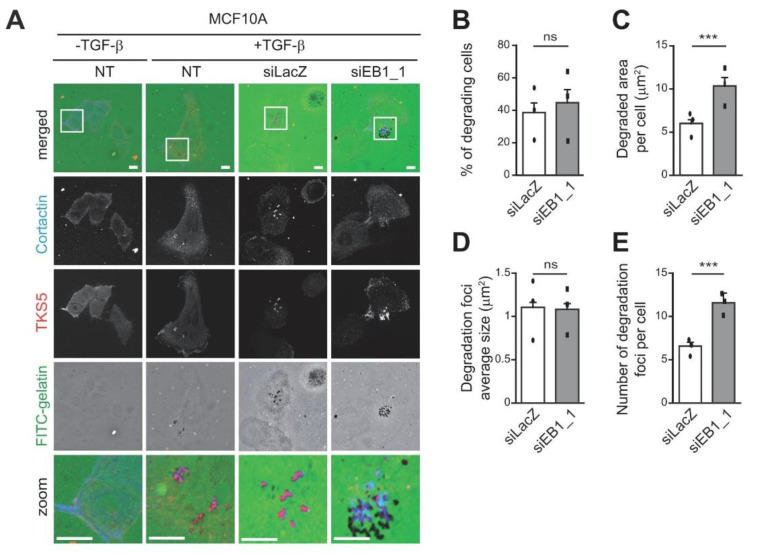
EB1 restricts ECM degradation via inhibition of invadopodia formation in TGF-β-transformed MCF10A breast epithelial cells. Non-transfected (NT) MCF10A cells or cells transfected with a control siRNA (siLacZ) or a siRNA against EB1 (siEB1_1) were treated or not with TGF-β for 6 days before seeding on FITC-gelatin for 4 h. Invadopodia and matrix degradation were identified as described in Figure 1. (**A**) Representative images are shown. The white-boxed regions are enlarged at the bottom (zoom). Scale bars represent 10 μm in non-enlarged images, 5 μm in enlarged images. (**B**–**E**) The ability of MCF10A cells to degrade fluorescently-labeled gelatin was analyzed. The percentage of degrading cells (**B**), the degraded area per cell (**C**), the average size of degradation foci (**D**), and the number of degradation foci (**E**) are represented as the mean ± SEM of three independent experiments. The mean of each individual experiment is reported. Statistical analysis was performed as described in Figure 1. *** *p* ≤ 0.001, ns not significant.

**Figure 3 cells-10-00388-f003:**
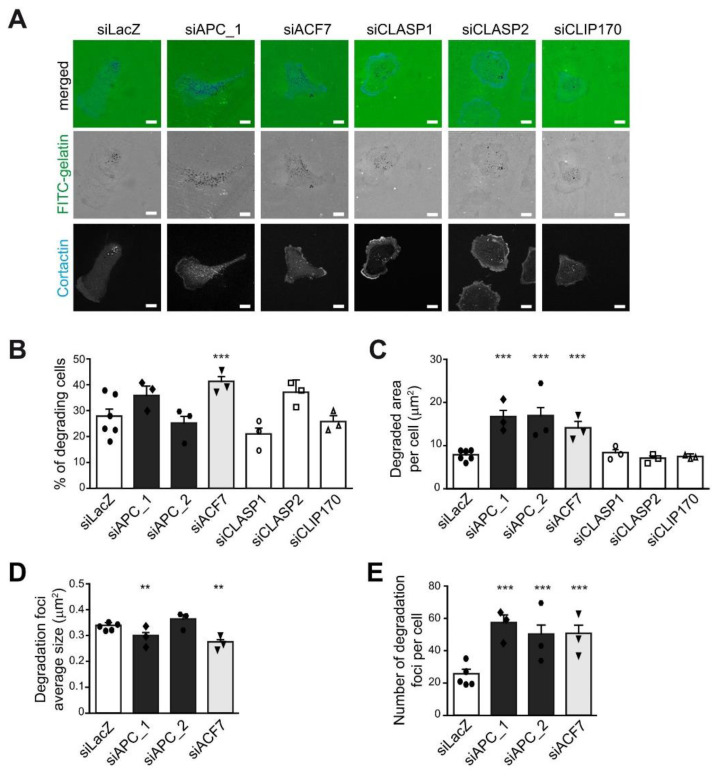
APC and ACF7, similarly to EB1, restricted ECM proteolysis by restraining invadopodia formation. MDA-MB-231 cells were transfected with a control siRNA control (siLacZ) or siRNA against APC (siAPC_1, siAPC_2), ACF7 (siACF7), CLASP1 (siCLASP1), CLASP2 (siCLASP2), and CLIP170 (siCLIP170) and seeded on fluorescently-labeled gelatin (FITC-gelatin) for 4 h. Cells were fixed and stained with an anti-Cortactin antibody to identify cell boundaries and invadopodia. Matrix degradation was identified thanks to the appearance of dark spots in FITC-gelatin. (**A**) Representative images are shown. Scale bars represent 10 μm. The ability of MDA-MB-231 cells described in (**A**) to degrade fluorescently-labeled gelatin was analyzed. The percentage of degrading cells (**B**), the degraded area per cell (**C**), the average size of degradation foci (**D**), and the number of degradation foci (**E**) are represented as the mean ± SEM of three independent experiments. The mean of each individual experiment is reported. Statistical analysis was performed as described in Figure 1. *** *p* ≤ 0.001, ** *p* ≤ 0.01.

**Figure 4 cells-10-00388-f004:**
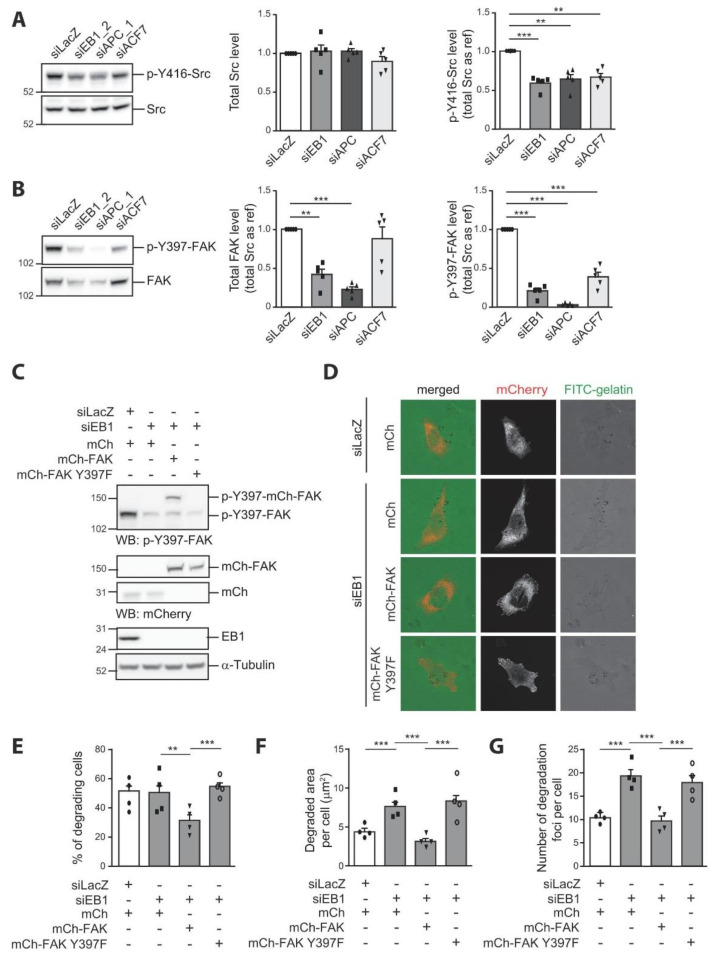
EB1 restricts invadopodia formation via inhibition of FAK activity. (**A**,**B**) MDA-MB-231 cells were transfected with a control siRNA (siLacZ) or siRNA directed against EB1 (siEB1_2), APC (siAPC_1) and ACF7 (siACF7). Levels of Src, activated Src (p-Y416-Src), FAK and FAK phosphorylated on Tyr397 (p-Y397-FAK) were analyzed by Western blotting using specific antibodies. (**A**) Total Src and p-Y416-Src levels are represented as the mean ± SEM of five independent experiments. The value of each individual experiment is reported. A representative Western blot is shown (left). (**B**) Total FAK and p-Y397-FAK levels are represented as the mean ± SEM of five independent experiments. The value of each individual experiment is reported. A representative Western blot is shown (left). Src was used as a loading control and as a reference to quantify p-Y416-Src, total FAK, and p-Y397-FAK levels. The paired *t*-test was used to determine significant differences. *** *p* ≤ 0.001, ** *p* ≤ 0.01. (**C**–**G**) MDA-MB-231 cells stably expressing mCherrry, mCherry fused to wild type FAK or mCherry fused to the SH2-binding motif mutant of FAK (FAK Y397F) were transfected with a control siRNA (siLacZ) or a siRNA against EB1 (siEB1) before seeding on fluorescently-labeled gelatin (FITC-gelatin) for 4 h. (**C**) Levels of p-Y397-FAK, mCherry fusion proteins, and EB1 were analyzed by Western blotting (WB) using specific antibodies. α-Tubulin serves as a loading control. (**D**) Representative images of cells fixed and stained with anti-mCherry and anti-Cortactin (not shown) antibodies. The percentage of mCherry positive degrading cells (**E**), the degraded area per cell (**F**), and the number of degradation foci (**G**) are represented as the mean ± SEM of four independent experiments. The mean of each individual experiment is reported. Statistical analysis was performed as described in Figure 1. Full-length blots are presented in Appendix A.

## Data Availability

Data presented in this study are contained within this article and the Appendix A, or available upon request to the corresponding author.

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
