# Peer review of "EB1 Restricts Breast Cancer Cell Invadopodia Formation and Matrix Proteolysis via FAK"

_cells, 2021, doi:10.3390/cells10020388_

Round 1
Reviewer 1 Report
The authors addressed the function of +Tips of microtubules in invadopodia formation. Their data suggest that EB1 restricts breast cancer cell invadopodia formation and matrix proteolysis via FAK. This is a nice study with all appropriate controls.
Would be nice if the authors could discuss the possible reasons for difference in effect of ACF7 vs. EB1 and APC on FAK (effect on (total, probably not relative) activity only in ACF7 vs. effect on FAK expresssion and activity in EB1 and APC.
Would be also nice to test the effect of EB1, APC and ACF7 knockout on breast cancer cell invasion, but not necessary from my point of view.
Reviewer 2 Report
In this manuscript, Chanez et al. showed the role of EB1, a major hub of the +TIP network, and its two additional partners APC and ACF7, in restricting invadopodia formation regulating FAK/Src activity, and in this way impacting the cancer-related processes. Knockdown of EB1, APC, and ACF7 increase the ECM degradation by invadopodia. The work provides an interesting mechanistic insight into the ability of EB1 in restricting ECM degradation hence cancer cell invasion, by regulating the FAK activity. I have a few questions and suggestions for the authors as below to strengthen the work and make it even better, confident and impactful.
1. My biggest concern is the no rescue experiments for the siRNAs. We know the siRNAs have off-target effects and in this study, authors home designed the siRNAs which were not used in any previous study. Rescue experiment just for a couple of important findings would make the results concrete. e.g., for figure 1. There are different ways authors can do it. They can use the EB1 construct for the experiments after doing the silencing. e.g., 17234 from addgene can be ordered if they already do not have. Then, either they use the siRNA, which targets the 3'-UTR, or if the siRNAs they used in the manuscript target the CDS, they can do silent mutation at the siRNA binding site in the EB1 construct. Fig 4 from Lilja et al. JCB 2017 is a good example of this strategy.
2. Did the authors try to double silence EB1 with APC or ACF7 to see whether the effects are additive? This could be a very insightful experiment.
3. A cartoon summary would be beneficial for the readers. e.g., something like in Franceschi et al. JCS 2015 review article.
4. Did the authors try to do some major experiments in any other breast cancer invasive cell lines like SKBR3? It is always better though not necessary, to validate important findings like in this paper in more than one cell line.
5. Now, there is no discussion and conclusion heading, they are merged with results. They should be separated.
6. Limitations of this study and future directions should be mentioned in the discussion.
7. No concentration of the siRNAs are mentioned. They should be mentioned!
8. If possible, functional assays like migration and invasion upon loss of EB1/EB1+APC/EB1+ACF7 can be further informative, though not necessary to do for this manuscript.
